# OpenReview forum: "Understanding the Performance Gap in Preference Learning: A Dichotomy of RLHF and DPO"
_ICML.cc/2026/Conference — ICML 2026 regular_

### Official Review · Reviewer_pUGr · 2026-02-27

**Soundness:** 3
**Presentation:** 3
**Significance:** 3
**Originality:** 3
**Overall Recommendation:** 4
**Confidence:** 3

**Summary:**

In this paper, the authors study the performance gap between RLHF, DPO, and Online DPO, providing a theoretical analysis of the performance gap between the three approaches (under an *exact approximation* assumption) in four different scenarios: ($i$) neither the reward model nor the policy model are mis-specified, ($ii$) only the policy model is mis-specified, ($iii$) only the reward model is mis-specified, and ($iv$) both the reward model and policy model are mis-specified. In Section 4, the authors consider the realistic case in which data are limited, leading to estimation errors. Finally, in Section 5, the authors propose simple evaluation experiments.

**Compliance With Llm Reviewing Policy:**

Affirmed.

**Final Justification:**

This paper explores the difference between RLHF and DPO and provides interesting theoretical results on when and why one method beats the other under several mispecifications. This improves the understanding of the two approaches. Whether this is something that can be used to improve either of the two, it is unclear to me as of now.
This motivates my recommendation. Nonetheless, in my opinion, this paper could be considered for acceptance at ICML.

**Key Questions For Authors:**

I would like the authors to clarify the result of Theorem 2. In particular, it is unclear to me the meaning of DPO loss's gradient being equal to the gradient of a quantity that contains stop-gradient operators. Moreover, policy $\pi^{\textsf{s}}$ appears in Theorem 2 but is not used in the result, and the noises $\epsilon_{y, y'}$ are unclear.

Also, regarding Theorem 11, can the authors provide a clearer explanation of $\Lambda_1$ and $\Lambda_2$?

**Limitations:**

Yes

**Strengths And Weaknesses:**

Although I have not thoroughly checked the appendix, the proposed results seem reasonable. The theoretical results support the discussion, which in itself is logical, structured, and consistent.
Overall, the paper is well structured; however, there are some presentation details that could be improved:

1. The reward model class $\mathcal{F}$ is introduced as parametric, and the dimension $d_R$ is employed in results, yet no additional information on how $\mathcal{F}$ is provided;
2. Remark 1 does not seem to be a remark but rather a continuation of Definition 1;
3. In Theorem 11, one result is presented as an upper bound, whereas the other is presented as a lower bound, which seems rather uninformative for a comparison. Also, quantities $\Lambda_1$ and $\Lambda_2$ are not intuitively explained;
4. The axes, titles, and legends of the Figures in the main paper are unreadable;
5. [Minor] At Line 42, (Zhu et al., 2023) is presented as a "fundational work of preference learning". This seems to be somewhat of an overstatement since ($i$) the cited paper is itself based upon RLHF and ($ii$) a more precise citation for preference learning could be (Fünrkranz and Hüllermeier, 2003).

In my opinion, the theoretical analysis of the difference between RLHF and DPO in different scenarios may be relevant for the scientific community.
Finally, I appreciate the authors' discussion on the limitations of their contributions in the main paper.


Fürnkranz, J. and Hüllermeier, E., 2003, September. Pairwise preference learning and ranking. In European conference on machine learning (pp. 145-156). Berlin, Heidelberg: Springer Berlin Heidelberg.

---

> ### Author Rebuttal · Authors · 2026-03-28
>
> > [W1] The reward model class $\mathcal F$ is introduced as parametric, and the dimension $d_R$ is employed in results, yet no additional information on how $\mathcal F$ is provided;
>
> A: The dimesnsion of the reward model class can be understood as the number of parameters in the reward model. And in Section 4, we explicity define the reward model and policy model, so its dimension is explicity $\Theta(d)$.
>
> > [W2] Remark 1 does not seem to be a remark but rather a continuation of Definition 1;
>
> A: Thank you for pointing this out. We will include Remark 1 in Definition 1 in next version.
>
> > [W3] In Theorem 11, one result is presented as an upper bound, whereas the other is presented as a lower bound, which seems rather uninformative for a comparison. Also, quantities $\Lambda_1,\Lambda_2$ are not intuitively explained;
>
> A: Our goal of establishing theorem 11 is to show the advantage of RLHF over DPO. So we present "**upper bound** of the error of RLHF" and "**lower bound** of the error of DPO". And when we take $n=O(d)$ and take $\Lambda_1,\Lambda_2$ as constants, we can see that "**upper bound** of the error of RLHF" < "**lower bound** of the error of DPO", so the RLHF is statistically better than DPO.
>
> As for $\Lambda_1,\Lambda_2$, they are formally defined in Appendix C.11.5, determined by the max/min eigenvalue of the Gram matrix, which is averaged over data. These quantities only depend on data, and can be taken as constants.
>
> > [W4] The axes, titles, and legends of the Figures in the main paper are unreadable;
>
> A: Thank you for pointing this out. We would increase the scale and fontsize of Figure 1-3 and add more illustrations in next version.
>
> > [W5, Minor] At Line 42, (Zhu et al., 2023) is presented as a "fundational work of preference learning". This seems to be somewhat of an overstatement since the cited paper is itself based upon RLHF and a more precise citation for preference learning could be (Fünrkranz and Hüllermeier, 2003).
>
> A: Thank you for this suggestion. We will recontextualized the citation of [Zhu et al, 2023] as showing the theoretical limits of RLHF, and cite [Fünrkranz and Hüllermeier, 2003] as the foundational work of preference learning.
>
> > [Q1] I would like the authors to clarify the result of Theorem 2. In particular, it is unclear to me the meaning of DPO loss's gradient being equal to the gradient of a quantity that contains stop-gradient operators. Moreover, policy $\pi^{s}$ appears in Theorem 2 but is not used in the result, and the noises $\epsilon_{y,y'}$ are unclear.
>
> A:  We first clarify why "DPO loss's gradient being equal to the gradient of a quantity that contains stop-gradient operators". $sg(Z_\theta(x))$ means that its gradient is 0, and $y\sim sg(\pi_\theta(\cdot\vert x))$ means we just sample $y$ from that distribution and don't have to compute policy gradient. You can refer to the formulation of online DPO in Section 2, there is also a stop-gradient operator in its objective. Please see detailed derivation in Appendix C.2. If it's still unclear, please let us know.
>
> The design of PILAF sampler is to remove the first-order error. And if we use other sampler, it will introduce a first-order error. So the result of Theorem 2 only holds when $\pi^{s}$ is the PILAF sampler. $\epsilon_{y,y'}$ is the Lagrange remainder. We didn't show details in the main content for simplicity. We will explicitly explain it in next version.
>
> > [Q2] Also, regarding Theorem 11, can the authors provide a clearer explanation of the quantities?
>
> A: Please see our response to W3.

---

> > ### Author Rebuttal · Reviewer_pUGr · 2026-04-01
> >
> > The Authors have answered my questions.
> >
> > I am not fully convinced that $\Lambda_1$ is indeed a constant, as, looking (although quickly) at Appendix C.11.5 and combining it with Assumption 13, it seems like $\| \Sigma_{\mathcal{D}|^{-1/2} \|_2$ may depend on $\sqrt{d}$. Nonetheless, this does not change the correctness of the Theorem, so this is not a concern.
> >
> > I will maintain my score, given that, considering the other reviews and rebuttals, it seems to me that all Reviewers are quite aligned towards acceptance.

---

> > > ### Author Response · Authors · 2026-04-01
> > >
> > > We are delighted to hear that your concerns are adequately addressed! We thank the reviewer again for the invaluable feedback and insights.
> > >
> > > We agree that $\Lambda_1,\Lambda_2$ cannot be directly obtained as constants. To make our result more general, we didn't explicitly make specific assumptions. The quantity $\Vert\Sigma_{\mathcal D}^{-1/2}\Vert_2$  is also adopted in Section 4 of [1] without explicit bounds. We will add this discussion in next version.
> > >
> > > [1] Reward Model Learning vs. Direct Policy Optimization: A Comparative Analysis of Learning from Human Preferences. Nika et al. ICML 2024.

---

### Official Review · Reviewer_4e9J · 2026-03-02

**Soundness:** 2
**Presentation:** 2
**Significance:** 3
**Originality:** 3
**Overall Recommendation:** 4
**Confidence:** 3

**Summary:**

The paper investigates the differences between RLHF methods which use an explicit reward model (e.g. Policy-Gradient+KL penalty) and method which use an implicit reward model (DPO).
The paper first compares them under reward model misspecification, policy model misspecification, or both, under exact optimization.
It then considers the finite data setting, and finally moves on to a small-scale experimental validation

**Compliance With Llm Reviewing Policy:**

Affirmed.

**Final Justification:**

My main concerns about the chosen two-token setting have been addressed in the rebuttal, I thus increased my score 3->4.

I did not increase it further because the empirical validation is relatively weak, but this is okay for a work with a theoretical focus.

**Key Questions For Authors:**

* Why is the optimal parameter $\theta_t^*$ time dependent in Section 4? It seems more natural that the hidden-state $\phi(y_{0\dots t})$ includes any token dependency but the reward parameters remain fixed? Could you explain what the impact of this on the following theoretical analysis is?


Post rebuttal: 3->4

**Limitations:**

The assumptions made for the theoretical analysis and their choice is not sufficiently discussed.
Societal impact is sufficiently addressed.

**Strengths And Weaknesses:**

Strengths:
 * Presentation: The exact optimization setting is introduced clearly and the main results are interesting and intuitive, though not entirely surprising.
 * Significance: The comparison of direct alignment vs reward modeling (RM)+policy gradient(PG) is practically relevant, both methods are actively used and it is not clear which is superior when, making this a timely contribution.


Weaknesses:
 * Presentation: It would be nice to include more explicit proof sketches in the main text. While the text explains the intuitive reasons for why each method performs favorably nicely, the presented theoretical results feel somewhat disconnected. For example, Theorem 2 requires a bounded reward difference (an almost optimal reward?) but it's not clear why or when this is justified. Likewise, $\epsilon$ is explained in the appendix but appears without explanation in the main text.

 * Soundness: The approximate optimization setting is limited to a two-token case, it is not clear to me why this setting was chosen over a simple bandit setting and whether it is more informative. Further, assumptions are omitted in the main text and only stated but not discussed in the Appendix.

 * Presentation: The "dichotomy of RLHF and DPO" is a false dichotomy. DPO is a method to solve the RLHF problem setting [1]. This submission really compares two-stage Reward Modeling -> Policy Optimization vs direct alignment.  Further, referring to RM+PG / PPO as "online" while also comparing with "Online DPO" is somewhat misleading, as - unlike online DPO - RM+PG does not obtain additional preference data during training. For this reason, the finding that "Online DPO can outperform both DPO and RM+PG" is not very surprising.

 * Related Work: [2] has previously investigated the consequences of misspecification in the two-stage RM+PG setting and provides an empirical comparison with DPO, thus is relevant to both the exact optimization and approximate optimization setting and should be compared to.

 * Soundness: The experiments are conducted with very small models (774M parameters), small datasets (<10k samples), and untuned hyper-parameters. Particularly for RL a batch-size of 8 is used, which is very unusually small. However, the core of this paper is the theoretical analysis so I don't think this is a huge issue in itself.


[1] Rafailov et al. "Direct Preference Optimization: Your Language Model is Secretly a Reward Model", NeurIPS 2023
[2] Ackermann et al. "Off-Policy Corrected Reward Modeling for Reinforcement Learning from Human Feedback", COLM 2025

---

> ### Author Rebuttal · Authors · 2026-03-28
>
> > [W1] Presentation: It would be nice to include more explicit proof sketches in the main text.
>
> A: Thank you for pointing out this presentation issue. We will introduce more proof sketches in the extra page if accepted.
>
> Theorem 2 is to show that the optimization objective of Online DPO can be very close to RLHF under some ideal assumptions like bounded reward difference and realizability (i.e., the DPO policy model is close to optimal). It also shows that potential that they could differ much when the bounded reward difference is violated, i.e., the policy model is mis-specified or badly fitted, which is further discussed in Section 3. We will add more disscussions in Remark 2 in next version.
>
> $\epsilon$ is the Lagrange remainder. We didn't show details in the main content for simplicity. We will explicitly explain it in next version.
>
> > [W2] Soundness: The approximate optimization setting is limited to a two-token case.
>
> A: We would like to clarify that, compared with single-token setting, multi-token setting is more close to real world practice, because transformer is a autoregressive probabilistic model. The linear multi-token model is also adopted in Section 5.1 of [3], Section 5.1 of [4], and Section 2 of [5]. In single-token setting, there is no statistical difference between RM and DPO, as we show in the second paragraph in Section 4. But by simply adding one more token, we can find a statiscal advantage of RM over DPO. Therefore, we think studying the two-token case is necessary and more informative. Our main theorem (Theorem 10,11) doesn't extend to $\ge 3$-token case, for mathematical simplicity. But our intuition in the "Curse of value function" paragraph hold for all multi-token cases.
>
> We will add remarks on these assumptions in the main context, and include more discussions, in the next version.
>
> [3] Principled Reinforcement Learning with Human Feedback from Pairwise or K-wise Comparisons. Zhu et al. ICML 2023.
>
> [4] Is a Good Foundation Necessary for Efficient Reinforcement Learning? The Computational Role of the Base Model in Exploration. Foster et al. COLT 2025.
>
> [5] The Coverage Principle: How Pre-Training Enables Post-Training. Chen et al. ICLR 2026.
>
> > [W3] Presentation: The "dichotomy of RLHF and DPO" is a false dichotomy.
>
> A: Thank you for pointing out this terminology confusion. We would consider replacing "RLHF" with "two-stage RLHF" to make it clearer.
>
> In exact optimization setting, RM+PG can obtain infinite preference data during training, but the data are generated from a fixed distribution. As a comparison, the data of online DPO are generated in an online way. Then the advantage of online DPO over DPO and RM+PG is not that obvious. And we want to clarify that "Online DPO can outperform both DPO and RM+PG" doesn't always hold. We  show that online DPO underperforms RM+PG under policy mis-specification in Section 3.
>
> > [W4] Related Work: [2].
>
> A: This paper [2] is highly relevant to our Section 3, where we show that overoptimization of mis-specified reward model can make RLHF worse than DPO (Prop 5 and Prop 7). The difference is that [2] focus on how to fix this issue through off-policy correction, and we focus on a systematic understanding of the performance gap. We will cite [2] and include disscusions in the next version.
>
> > [W5] The experimental scale is small. However, the core of this paper is the theoretical analysis.
>
> A: We appreciate the reviewer’s recognition of our theoretical focus. Our primary contributions are theoretical, and the purpose of the experiments is to validate the qualitative predictions of our theoretical analysis, but not to serve as large-scale empirical benchmarks. We intentionally use controlled, small-scale setups for two reasons: (1) to keep experimental conditions interpretable and aligned with the assumptions in each theoretical scenario, and (2) to avoid obscuring the predicted effects with noise from large-scale training dynamics. To further validate our theoretical claims, we've also shown connections to existing large-scale experiments in Section 5.2.
>
> > [Q1] Why is the optimal parameter $\theta_t^\star$ time-dependent in Section 4? It seems more natural that the hidden-state includes any token dependency but the reward parameters remain fixed? Could you explain what the impact of this on the following theoretical analysis is?
>
> A: In the standard token-level linear model adopted in [3,4,5], the optimal parameter $\theta$ is time-independent, i.e., $\theta^\star_{t1}=\theta^\star_{t2}$, $\forall t_1,t_2$. Our time-dependent model is a more general version, which allows $\theta^\star_{t1},\theta^\star_{t2}$ to vary. Changing it to time-independent won't affect our claims in Section 4. This formulation is to ensure mathematical simplicity in proofs of Theorem 10,11, so that we can technically focus on the first-token as stated in Remark 5. Without this formulation, we can still prove the result, but that would be in a more complicated way.

---

> > ### Author Rebuttal · Reviewer_4e9J · 2026-04-01
> >
> > I appreciate the rebuttal provided by the authors.
> >
> > Particularly the discussion of the two-token setting and the time-dependence of $\theta$ has been helpful, I would recommend adding them to the paper.
> >
> > I am raising my score, but only to weak accept as I still have concerns about the limitations of the empirical evaluation.
> >
> > Nonetheless, I think this paper is useful to the community and should be accepted.

---

> > > ### Author Response · Authors · 2026-04-01
> > >
> > > We are delighted to hear that the reviewer has raised the score! We thank the reviewer again for the invaluable feedback and insights.

---

### Official Review · Reviewer_KZEH · 2026-03-06

**Soundness:** 3
**Presentation:** 3
**Significance:** 3
**Originality:** 3
**Overall Recommendation:** 4
**Confidence:** 4

**Summary:**

This paper theoretically analyzes the performance gap between RLHF and DPO (and online DPO). The authors first consider the case where optimization is exact, i.e., infinite samples and perfect optimization. Here, the authors study various cases of mis-specification, e.g., the case where the reward function is not realizable but the optimal policy is, and so on. In this case, various relationships (performance gaps) are derived. For example, in the aforementioned case, in a worst case, RLHF can suffer severely from the misspecification causing it to underperform compared to DPO. Other intuitive results are subsequently derived. The authors then consider the approximate optimization case, where they focus on a linear token-level parameterization and show that under certain conditions RLHF can be more sample-efficient.

**Compliance With Llm Reviewing Policy:**

Affirmed.

**Final Justification:**

I decided to keep my original, positive score. My main concern that it is unclear how much generality can be claimed remains after the rebuttal. This is concern is not unusual for work of this type, which derives "worst-case" results that rely on toy counterexamples.

**Key Questions For Authors:**

1. Many of the results rely on specific, simple bandit counterexamples to make statements about the relationship between DPO and RLHF. For example, Proposion 7-9. It is not clear to me whether the shown relationships are predictive of the behavior in practice. Do you think that your results actually extend / generalize beyond your simple MAB "counterexamples"? Why, why not? Please also comment on this point for the results in the approximate approximation setting in Section 4 (see below).
2. I haven't encountered the token-level linear parameterization formulation from Section 4 before. Is it a common formulation? In practice, reward models are not evaluated on a per-token level. What is the motivation to do it here (beyond for the sake of doing something different than previous work)?
3. Related to the above two points, you write in line 364 in Section 4 that
> "Although the parameterization is specific, it reveals a general phenomenon: DPO can distort the intrinsic structure of the true reward function. For general model class beyond linear model, Equation (1) still holds. And hence, compared with the reward model, the policy model always faces a more complext target, ..."

But to me the results appear specific to your modeling / parameterization. Can you really claim the generality here?
Also, everything after "And hence, ..." seems to me like intuition and not fact, as your results require specific assumptions. (I'm not arguing against the intuition here, in fact I agree with it, but you make seem as proven fact.)


---

Also: You should increase the fontsize in your Figure 1-3. It is impossible to read without zooming in / when printing the paper.

**Limitations:**

yes

**Strengths And Weaknesses:**

### Strenghts
1. Understanding the performance gap between RLHF and DPO from a theoretical perspective is an interesting and relevant problem to study.
2. The paper is well-written, easy to follow and I particularly appreciate that the results are properly contextualized within the existing literature on RLHF vs DPO (vs online DPO).


### Weaknesses
1. Despite the generality with which the results are presented, it is not clear whether they in fact are as general as they seem. Many results rely on specific multi-armed bandit examples (e.g., Prop. 8), and it is unclear whether this really supports a broader claim about RLHF vs DPO. For this, please refer to the questions below.
2. A minor weakness is that the experiments are conducted for very small and old models (GPT-2 774M). However, this is not a serious concern for me as the main contributions of this work are theoretical and the authors did a good job outlining other empirical results that support their hypotheses in the existing literature in Section 5.2.

Overall, the paper makes interesting contributions to better understand RLHF vs DPO. While one can argue that these are incremental or do not generalize beyond the worst case examples used in their proofs, I lean towards acceptance for now and gladly discuss in more depth with the authors.

---

> ### Author Rebuttal · Authors · 2026-03-28
>
> > [W2] A minor weakness is that the experiments are conducted for very small and old models (GPT-2 774M). However, this is not a serious concern for me as the main contributions of this work are theoretical and the authors did a good job outlining other empirical results that support their hypotheses in the existing literature in Section 5.2.
>
> We appreciate the reviewer’s recognition of our theoretical focus and the complementary evidence presented in Section 5.2. Our primary contributions are theoretical, and the purpose of the experiments is to validate the qualitative predictions of our theoretical analysis, but not to serve as large-scale empirical benchmarks. We intentionally use controlled, small-scale setups for two reasons: (1) to keep experimental conditions interpretable and aligned with the assumptions in each theoretical scenario, and (2) to avoid obscuring the predicted effects with noise from large-scale training dynamics.
>
> > [Q1] Many of the results rely on specific, simple bandit counterexamples to make statements about the relationship between DPO and RLHF. For example, Proposion 7-9. It is not clear to me whether the shown relationships are predictive of the behavior in practice. Do you think that your results actually extend / generalize beyond your simple MAB "counterexamples"? Why, why not? Please also comment on this point for the results in the approximate approximation setting in Section 4 (see below).
>
> A: Our counterexamples serve as existence proofs, so most of them are based on toy examples (like simple bandits) for mathematical rigor, but they can still have implications for practice.
> - Proposition 7 shows that online DPO has the potential to outperform RLHF when the reward model and policy model are equally strong, and in practice, we can view online DPO as actively updating the reward model, while RLHF is to use a fixed reward model, so that online DPO has the potential to outperform RLHF;
> - Proposition 8 and 9 indicate that under complex mis-specification scenarios, there is no explicit separation between DPO and RLHF, and in practice, we have that DPO can circumvent the reward model mis-specification but the optimization objective is a proxy, while RLHF can optimize the real objective while suffering from overoptimization of reward model mis-specification, so there is a trade-off and they can outperform each other in certain settings.
>
> As for Section 4, please see our response to Q2.
>
> > [Q2] I haven't encountered the token-level linear parameterization formulation from Section 4 before. Is it a common formulation? In practice, reward models are not evaluated on a per-token level. What is the motivation to do it here (beyond for the sake of doing something different than previous work)?
>
> A: The token-level linear parameterization is a natural extension of the linear softmax model, and this formulation is adopted in Section 5.1 of [1], Section 5.1 of [2], and Section 2 of [3].
>
> The token-level reward model formulation is a stronger version of the linear reward model $r_\theta(y)=\beta\theta^\top\psi(y)$, and it's designed for fair comparison with policy model in statistics by aligning the parameter counts. It's also adopted in Section 5.1 of [1]. And in practice, it is still an appropriate formulation, since **we can only observe the trajectory-level aggregated rewards**.
>
> [1] Principled Reinforcement Learning with Human Feedback from Pairwise or K-wise Comparisons. Zhu et al. ICML 2023.
>
> [2] Is a Good Foundation Necessary for Efficient Reinforcement Learning? The Computational Role of the Base Model in Exploration. Foster et al. COLT 2025.
>
> [3] The Coverage Principle: How Pre-Training Enables Post-Training. Chen et al. ICLR 2026.
>
> > [Q3] But to me the results appear specific to your modeling / parameterization. Can you really claim the generality here? Also, everything after "And hence, ..." seems to me like intuition and not fact, as your results require specific assumptions. (I'm not arguing against the intuition here, in fact I agree with it, but you make seem as proven fact.)
>
> A: As proved in Appendix C.10, Equation (1) holds for any autoregressive parameterization. So it's generally true that RM is estimating the reward function while DPO is estimating the Q function. And after "And hence", we mean that Q function is more complicated than reward function, so it's harder for DPO to accurately estimate it. We agree that "Q function is more complicated than reward function" is intuitive rather than rigorous (in [4] they attribute it to the verification-generation gap), and will further clarify this point in next version.
>
> [4] All Roads Lead to Likelihood: The Value of RL in Fine-Tuning. Swamy et al. ICLR 2026.
>
> > [Q4] Also: You should increase the fontsize in your Figure 1-3. It is impossible to read without zooming in / when printing the paper.
>
> A: Thank you for pointing this out. We will increase the scale and fontsize of Figure 1-3 in next version.

---

> > ### Author Rebuttal · Reviewer_KZEH · 2026-04-01
> >
> > Thank you for your detailed response.
> >
> > I decided to keep my current score, which is leaning towards acceptance.
> >
> > My main concern that it is very much unclear how much generality can be claimed remains. You have worst case bounds that rely on toy counter examples. It is to a certain degree inherent to the worst-case approach taken in the paper.

---

> > > ### Author Response · Authors · 2026-04-01
> > >
> > > Thank you for keeping your positive rating and providing insightful feedbacks.
> > >
> > > We'd like to further clarify the insights of our results here.
> > > - For Prop 7, we show an example where online DPO > RLHF. Recall Prop 6 where we proved RLHF=DPO. So we only need to construct online DPO > DPO. And in practice, "online DPO > DPO" is widely validated, such as in [1,2].
> > > - For Prop 8,9, we show examples where DPO and RLHF can outperform each other. And in practice we have:
> > >   - Figure 2 of [3] demonstrates that DPO can beat PPO under a weak reward model.
> > >   - Table 5 of [4] demonstrates that RLHF can beat DPO under a good reward model.
> > >
> > > Therefore, our results are not restricted to worst-case toy examples, and have practical implications.
> > >
> > > [1] Iterative preference learning from human feedback: Bridging theory and practice for rlhf under kl-constraint. Xiong et al. ICML 2024.
> > >
> > > [2] Preference Fine-Tuning of LLMs Should Leverage Suboptimal, On-Policy Data. Tajwar et al.
> > >
> > > [3] Your language model is secretly a reward model. Rafailov et al. NeurIPS 2023.
> > >
> > > [4] Unpacking DPO and PPO: Disentangling best practices for learning from preference feedback. Ivison et al. NeurIPS 2024.

---

### Official Review · Reviewer_Cca6 · 2026-03-11

**Soundness:** 3
**Presentation:** 3
**Significance:** 3
**Originality:** 3
**Overall Recommendation:** 5
**Confidence:** 3

**Summary:**

The paper studies when and why RLHF outperforms DPO (and vice versa) by analyzing two sources of performance gap. First, under exact optimization (infinite data), the authors set up a 2×2 taxonomy over reward-model and policy-model realizability, showing that the relative performance depends on which model class is mis-specified. Second, under finite samples, they construct a dual-token sparse prediction (DTSP) task where RLHF's reward learning achieves $\tilde{O}(\sqrt{k \log d / n})$ estimation error while DPO is stuck at $\Omega(d/n)$, owing to the fact that DPO must implicitly learn a value function that destroys the sparsity structure of the true reward.

**Compliance With Llm Reviewing Policy:**

Affirmed.

**Final Justification:**

I maintain my score. This is solid work.

**Key Questions For Authors:**

1. Can you provide any evidence that real-world rewards are sparse in a meaningful feature basis (empirical or theoretical)?
2. How sensitive is the second-order error in Theorem 2 to the choice of sampler?
3. It would be worth briefly discussing whether the results extend beyond the Bradley-Terry preference model.

**Limitations:**

yes

**Strengths And Weaknesses:**

## Strengths

- The 2×2 taxonomy in Table 1 is useful. It gives a clean decision framework, and I think practitioners will find this helpful. The isomorphic double-mis-specification result (Proposition 7), where online DPO can strictly beat both RLHF and offline DPO, is the most interesting finding in Section 3 and was not obvious to me before reading.

- The "curse of value function" is the paper's best idea. Comparing Equations (3) and (4) makes it immediately clear why DPO has a harder statistical problem. This also explains intuitively why sparsity in the reward does not help DPO.

- The connections to existing large-scale experiments are well done. Given that the paper's own experiments are small-scale, these connections do a lot of work in making the theoretical claims feel grounded.

## Weaknesses

- Theorem 10 requires rewards to be sparse ($k \ll d$), but the paper does not show why this holds for real tasks. Without this, the finite-sample story collapses to "RLHF and DPO have comparable sample complexity," which is exactly what the linear bandit case already shows.

-  Propositions 1, 3, 5, and 6 are either known or direct consequences. Propositions 4 and 7 are not actually proved.

---

> ### Author Rebuttal · Authors · 2026-03-28
>
> > [W1] Theorem 10 requires rewards to be sparse ($k\ll d$), but the paper does not show why this holds for real tasks. Without this, the finite-sample story collapses to "RLHF and DPO have comparable sample complexity," which is exactly what the linear bandit case already shows.
>
> A: Sparsity is a common phenomenon in preference learning, particularly in high-dimensional feature spaces where human preferences are often driven by only a small subset of factors. For examples, people's preference over smart phones could depend on a few factors like price, camera quality and battery life; a reader’s preference over responses may be determined by the presence of only a few key words; and there are many recommender systems paper relying on "only a few factors determine a user’s preference for a movie" [1]. Therefore, in a high-dimensional preference model, the corresponding reward parameter is naturally sparse. This aspect is also discussed in [2]. In Section 4.2 of [2], they empirially report that given a pre-trained reward model backbone with frozen features $\psi$, a effective fine-tuned reward model can have $k/d$ equal to $4.2$%-$7.5$%.
>
> Beyond sparsity, in practice we can also look into other properties such as low-rank structures, which are proved to exist in real data [3,4]. As we discussed in the paragraph "Curse of value function", our insights for practice is that value function will distort the simple structure of reward function and makes it harder to learn.
>
> [1] Matrix Factorization Techniques for Recommender Systems. Koren et al. Computer, 42(8), 30-37.
>
> [2] Leveraging Sparsity for Sample-Efficient Preference Learning: A Theoretical Perspective. Yao et al. ICML 2025.
>
> [3] Intrinsic Dimensionality Explains the Effectiveness of Language Model Fine-Tuning. Aghajanyan et al. ACL 2021.
>
> [4] Why Are Big Data Matrices Approximately Low Rank? Udell et al. SIAM Journal on Mathematics of Data Science.
>
> > [W2] Propositions 1, 3, 5, and 6 are either known or direct consequences. Propositions 4 and 7 are not actually proved.
>
> A: We agree that some of the results are already established (Prop 1,6) or direct (Prop 3,5), but we included them for a systematic taxonomy, and these propositions are simple yet useful (as shown in Section 5.2). Also, their conciseness is a deliberate result of our proposed definitions of different conditions, which allow these properties to be derived naturally.
>
> We will formalize the numerical construction in the proofs of proposition 4,7. For the proof of proposition 4, we only need to show that the gradient of DPO/online DPO w.r.t. $x$ is positive when $x\in (0,4]$ and negative when $x\in [-4,0)$; For the proof of proposition 7, we only need to show that the gradient of online DPO w.r.t. $x$ is increasing and has a zero point $x_0\in[-4,4]$, while the gradient of DPO w.r.t. $x$ is negative for $x\in[-4,4]$. These are all simple arithmetics. We will include the formal proofs in next version.
>
> > [Q1] Can you provide any evidence that real-world rewards are sparse in a meaningful feature basis (empirical or theoretical)?
>
> A: Please see our response to W1.
>
> > [Q2] How sensitive is the second-order error in Theorem 2 to the choice of sampler?
>
> A: The second-order error comes from Lagrange remainder, and is controlled as long as the rewards and the reward differences are bounded. We would like to clarify that the design of PILAF sampler is to remove the first-order error. And if we use other sampler, it will also introduce a first-order error.
>
> > [Q3] It would be worth briefly discussing whether the results extend beyond the Bradley-Terry preference model.
>
> A: The design of DPO is baed on the Bradley-Terry preference model, so extension to other preference model requires modifications to objectives of DPO and reward modeling. Our general results in Section 3 can generalize to random utility model such as Thurstone-Mosteller model [5,6], where $P(y_w\succ y_l)=F(\frac{r(y_w)-r(y_l)}{\sigma})$. It's also worth exploring more general setting such as Nash learning. We will include this discussion in next version.
>
> [5] Thurstone, L. L. A law of comparative judgment. Psychological review, 101(2):266, 1994.
>
> [6] Mosteller, F. Remarks on the method of paired comparisons: I. the least squares solution assuming equal standard deviations and equal correlations. Selected Papers of Frederick Mosteller, pp. 157–162, 2006.

---

> > ### Author Rebuttal · Reviewer_Cca6 · 2026-04-01
> >
> > My concerns have been addressed.

---

> > > ### Author Response · Authors · 2026-04-01
> > >
> > > We are delighted to hear that your concerns are adequately addressed! We thank the reviewer again for the invaluable feedback and insights.

---

### Decision · Program_Chairs · 2026-04-30

**Decision:**

Accept (regular)

**Comment:**

This paper theoretically studies the performance gap between RLHF and DPO (including online DPO). In the exact optimization setting, the authors analyze several misspecification regimes and derive performance comparisons, showing cases where RLHF can significantly underperform DPO under worst-case misspecification. They then extend the analysis to approximate optimization with a linear token-level parameterization, where RLHF can be more sample-efficient under certain conditions.

Most reviewers agreed that the paper merits acceptance, and after carefully reviewing the rebuttal and discussion, I share this view. That said, I recommend that the authors incorporate the reviewers’ feedback into the final version of the paper, with particular attention to:

1- Some part of the theoretical results are known or not proved.

2- The experimental results are on the old models and is not comprehensive

3- Some related works are missing and is not discussed in the paper